# Robust Multi-Agent Reinforcement Learning via Adversarial Regularization: Theoretical Foundation and Stable Algorithms

**Alexander Bukharin**[†]  **Yan Li**[†]  **Yue Yu**[†]  **Qingru Zhang**[†]  **Zhehui Chen**[‡]  **Simiao Zuo**[§]

**Chao Zhang**[†]  **Songan Zhang**[¶]  **Tuo Zhao**[†]

## Abstract

Multi-Agent Reinforcement Learning (MARL) has shown promising results across several domains. Despite this promise, MARL policies often lack robustness and are therefore sensitive to small changes in their environment. This presents a serious concern for the real world deployment of MARL algorithms, where the testing environment may slightly differ from the training environment. In this work we show that we can gain robustness by controlling a policy's Lipschitz constant, and under mild conditions, establish the existence of a Lipschitz and close-to-optimal policy. Based on these insights, we propose a new robust MARL framework, ERNIE, that promotes the Lipschitz continuity of the policies with respect to the state observations and actions by adversarial regularization. The ERNIE framework provides robustness against noisy observations, changing transition dynamics, and malicious actions of agents. However, ERNIE's adversarial regularization may introduce some training instability. To reduce this instability, we reformulate adversarial regularization as a Stackelberg game. We demonstrate the effectiveness of the proposed framework with extensive experiments in traffic light control and particle environments. In addition, we extend ERNIE to mean-field MARL with a formulation based on distributionally robust optimization that outperforms its non-robust counterpart and is of independent interest. Our code is available at `https://github.com/abukharin3/ERNIE`.

## 1 Introduction

In the past decade advances in deep neural networks and greater computational power have led to great successes for Multi-Agent Reinforcement Learning (MARL), which has achieved success on a wide variety of multi-agent decision-making tasks ranging from traffic light control [1] to StarCraft [2]. However, while much effort has been devoted to applying MARL to new problems, there has been limited work regarding the robustness of MARL policies.

Despite the limited attention paid to robustness, it is essential for MARL policies to be robust. Most MARL policies are trained in a fixed environment. Since these policies are trained solely to perform well in that environment, they may perform poorly in an environment with slightly different transition dynamics than the training environment. In addition, while agents are fed with exact state information in training, MARL policies deployed in the real world can receive inaccurate state information (e.g., due to sensor error). Finally, even a single agent acting maliciously or differently than expected can cause a chain reaction that destabilizes the whole system. These phenomena cause significant concern for the real-world deployment of MARL algorithms, where the environment dynamics and observation noise can change over time. We observe that even when the change in the environment's dynamics is small, the performance of MARL algorithms can deteriorate severely (See an example

---

[†] Georgia Institute of Technology    [‡] Google    [§] Microsoft    [¶] Ford Motor Company

37th Conference on Neural Information Processing Systems (NeurIPS 2023).

in Section 5). Thus there is an emerging need for MARL algorithms that are robust to changing transition dynamics, observation noise, and changing behavior of agents.

Although many robust RL methods have been proposed for the single agent case, three major barriers prevent their use for MARL. Theoretically, it is not clear if or when such methods can work for MARL. Methodologically, it is not straightforward to apply single agent robust RL methods to MARL, as single agent methods may not consider the interactions between several agents. Algorithmically, single agent robust RL algorithms are often unstable, and may not perform well when applied to inherently unstable MARL training. Therefore to learn robust MARL policies, we provide theoretical, methodological, and algorithmic contributions.

**Theory.** Theoretically, we first show that when the transition and reward function are smooth, a policy's value function is also smooth. In our experiments, we show that this assumption can serve as a useful prior knowledge, even if the transition function is not smooth in every state. Second, we prove that a smooth and close-to-optimal policy exists in any such environment. Third, we show that a policy's robustness is proportional to its Lipchitz constant *with no smoothness assumption on the environment's smoothness*. These observations advocate for using smoothness as an inductive bias to not only reduce the policy search space, but simultaneously improve the robustness of the learned policy. Finally, we prove that large neural networks are capable of approximating the target policy or Q functions with smoothness guarantees. These findings give us the key insight that in order to learn robust and high-performing deep MARL policies, we should enforce the policies' smoothness.

**Method.** Based on these findings, we propose a new training framework – advErsarially Regularized multiageNt reInforcement lEarning (ERNIE), that applies adversarial training to learn smooth and robust MARL policies in a principled manner. In particular, we develop an adversarial regularizer to minimize the discrepancy between each policy's output given a perturbed observation and a non-perturbed observation. This adversarial regularization gives two main benefits: Lipschitz continuity and rich data augmentation with adversarial examples. The adversarial regularization encourages the learned policies to be Lipschitz continuous, improving robustness. Augmenting the data with adversarial examples further provides robustness against environment changes. Specifically, new scenarios emerge when the environment changes, and data augmentation with adversarial examples provides a large coverage of these scenarios as long as the environment change is small. Adapting to adversarial examples during training ensures that the agents will perform reasonably even in the worst case.

To further provide robustness against the changing behaviors of a few malicious agents, we propose an extension of ERNIE that minimizes the discrepancy between the global Q-function with maliciously perturbed joint actions and non-perturbed joint actions. This regularizer encourages the policies to produce stable outputs even when a subset of agents acts sub-optimally, therefore granting robustness. Such robustness has not been considered in previous works.

**Algorithm.** We find that adversarial regularization can improve robust performance [3]. However, adversarial regularization can also be unstable. More concretely, conventional adversarial regularization can be formulated as a zero-sum game where the defender (the policy) and attacker (the perturbation) hold equal positions and play against each other. In this case, a small change in the attacker's strategy may result in a large change for the defender, rendering the problem ill-conditioned. Coupled with the already existing stability issues that come with training MARL algorithms, this instability issue greatly reduces the power of adversarial regularization methods for MARL.

To address this issue, we reformulate adversarial training as a Stackelberg game. In a Stackelberg game, the leader (defender) has the advantage as it knows how the follower (attacker) will react to its actions and can act accordingly. This advantage essentially makes the optimization problem smoother for the defender, leading to a more stable training process.

**Extension to Mean-field MARL.** We further demonstrate the general applicability of ERNIE by developing its extension to robustify mean-field MARL algorithms. The mean-field approximation has been widely received as a practical strategy to scale up MARL algorithms while avoiding the curse of many agents [4]. However, as mean-field algorithms are applied to real-world problems, it is essential to develop robust versions. To facilitate policy learning that is more robust, we extend ERNIE to mean-field MARL with a formulation based on distributionally robust optimization [5; 6].

To demonstrate the effectiveness of the proposed framework, we conduct extensive experiments that evaluate the robustness of ERNIE on traffic light control and particle environment tasks. Specifically,

we evaluate the robustness of MARL policies when the evaluation environment deviates from the training environment. These deviations include observation noise, changing transition dynamics, and malicious agent actions. The results show that while state-of-the-art MARL algorithms are sensitive to small changes in their environment, the ERNIE framework enhances the robustness of these algorithms without sacrificing efficiency.

**Contributions.** We remark that adversarial regularization has been developed for single-agent RL, but never for MARL [3]. Our contribution in this paper has four aspects: (1) advances in theoretical understanding (2) development of new regularizers for MARL (3) new algorithms for stable adversarial regularization in MARL (4) comprehensive experiments in a number of environments.

## 2  Background

In this section, we introduce the necessary background for MARL problems together with related literature. We consider the setting of *cooperative MARL*, where agents work together to maximize a global reward.

- **Cooperative Markov Games**. We consider a partially observable Markov game $\langle \mathcal{S}, \mathcal{O}^N, \mathcal{A}^N, \mathcal{P}, \mathcal{R}, N, \gamma \rangle$ in which a set of agents interact within a common environment. We let $\mathcal{S} \subseteq \mathbb{R}^S$ denote the global state space, $\mathcal{O} \subseteq \mathbb{R}^O$ denote the observation space for each of the $N$ agents, $\mathcal{A} \subseteq \mathbb{R}^A$ denote the action space, $\mathcal{P} : \mathcal{S} \times \mathcal{A} \mapsto \mathcal{S}$ denote the transition kernel, $\gamma$ denotes the discount factor, and $\mathcal{R} : \mathcal{S} \times \mathcal{A} \mapsto \mathbb{R}^N$ denotes the reward function. At every time step $t$, each of the $N$ agents selects an action according to its policy, which can be stochastic or deterministic. Then, the system transitions to the next state according to the transition kernel and each agent receives a reward $r_{i,t}$. We denote the global reward at time $t$ as $r_t^g$. The goal of each agent is to find a policy that maximizes the discounted sum of its own reward, $\sum_{t \geq 0} \gamma^t r_{i,t}$.

- **Robust RL**. In recent years many single agent robust RL techniques have been proposed. Most of these methods use information about the underlying simulator to train agents over a variety of relevant environment settings [7; 8; 9; 10; 11]. Although these methods can provide robustness against a wide range of environment changes, they suffer from long training times and require expert knowledge of the underlying simulator, which is not practical. Another direction of research focuses on perturbation based methods [3; 12]. Perturbation based methods train the policy to be robust to input perturbations, encouraging the policy to act reasonably in perturbed or previously unseen states. [13] certify robustness by adding smoothing noise to the state; it is not clear how this affects the learned policy's optimality. Another related line of work [14; 15; 16; 17] studies robust markov decision processes and provides a principled way to learn robust policies. However, such methods often require strict assumptions on the perturbation/uncertainty. Inspiring our work, [3] proposes to learn a smooth policy in single agent RL, but they do so to reduce training complexity rather than increase robustness and provide no theoretical justification for their method. Instead, we theoretically connect smoothness to robustness, extend perturbation based methods to MARL, and develop a more stable perturbation computation technique, and develop an extension to mean-field MARL.

- **Robust MARL**. Recently, some works have studied the robustness of MARL systems. Lin et al. [18] studies how to attack MARL systems and finds that MARL systems are vulnerable to attacks on even a single agent. Zhang et al. [19] develop a framework to handle MARL with model uncertainty by formulating MARL as a robust Markov game. However, their proposed method only considers uncertainty in the reward function, while this article focuses on robustness to observation noise and changing transition dynamics. Li et al. [20] modify the MADDPG algorithm to consider the worst-case actions of the other agents in continuous action spaces with the M3DDPG algorithm. M3DDPG aims to grant robustness against the actions of other agents, which is less general than the robustness against observation noise, changing transition dynamics, and malicious agents that our method aims for. Wang et al. [21] consider robustness against uncertain transition dynamics, but their algorithm is not applied to deep MARL. More recently, He et al. [22]; Han et al. [23] introduces the concept of robust equilibrium and proposes to learn an adversarial policy to perturb each agent's observations. Finally Zhou et al. [24] propose to learn robust policies by minimizing the cross-entropy loss between agent's actions in non-perturbed states and perturbed states.

The ERNIE framework is also related to several existing works which use similar adversarial training methods but target different domains such as trajectory optimization [25], semi-supervised learning [26; 27; 28], fine-tuning language models [29; 30], and generalization in supervised learning [31].

# 3 From Lipschitz Continuity to Robustness

This section presents the theoretical motivation for our algorithm by showing that Lipschitzness (smoothness) serves as a natural way to gain robustness, while reducing the policy search space. We start by observing that certain natural environments exhibit smooth transition and reward functions, especially when the transition dynamics are governed by physical laws (e.g., MuJuCo environment [32], Pendulum [33]).* Formally, this is stated as the following.

**Definition 3.1.** Let $\mathcal{S} \subseteq \mathbb{R}^d$. We say the environment is $(L_r, L_\mathbb{P})$-smooth, if the reward function $r : \mathcal{S} \times \mathcal{A} \to \mathbb{R}$, and the transition kernel $\mathbb{P} : \mathcal{S} \times \mathcal{S} \times \mathcal{R}$ satisfy

$$|r(s,a) - r(s',a)| \le L_r \|s - s'\| \quad \text{and} \quad \|\mathbb{P}(\cdot|s,a) - \mathbb{P}(\cdot|s',a)\|_1 \le L_\mathbb{P} \|s - s'\|,$$

for $(s, s', a) \in \mathcal{S} \times \mathcal{S} \times \mathcal{A}$. $\|\cdot\|$ denotes a metric on $\mathbb{R}^d$. We say a policy $\pi$ is $L_\pi$-smooth if

$$\|\pi(\cdot|s) - \pi(\cdot|s')\|_1 \le L_\pi \|s - s'\|.$$

Without loss of generality, we assume $|r(s,a)| \le 1$ for any $(s, a) \in \mathcal{S} \times \mathcal{A}$. We then present our theory. Due to the space limit, we defer all technical details to the appendix.

• **From smooth environments to smooth values.** We proceed to show that if the environment is smooth, then the value functions for smooth policies are also smooth.

**Theorem 3.1.** *Suppose the environment is $(L_r, L_\mathbb{P})$-smooth. Then the Q-function of any policy $\pi$, defined as*

$$Q^\pi(s,a) = \mathbb{E}^\pi \left[ \sum_{t=0}^\infty \gamma^t r(s_t, a_t)|s_0 = s, a_0 = a \right], \ \forall(s,a),$$

*is Lipschitz continuous in the first argument. That is,*

$$|Q(s,a) - Q(s',a)| \le L_Q \|s - s'\|, \tag{1}$$

*where $L_Q := L_r + \gamma L_\mathbb{P}/(1 - \gamma)$. Suppose in addition the policy is $L_\pi$-smooth. Then the value function, defined as*

$$V^\pi(s) = \mathbb{E}^\pi \left[ \sum_{t=0}^\infty \gamma^t r(s_t, a_t)|s_0 = s \right], \ \forall s,$$

*is Lipschitz continuous. That is,*

$$|V^\pi(s) - V^\pi(s')| \le L_V \|s - s'\|,$$

*where $L_V := L_\pi/(1 - \gamma) + L_Q$.*

In view of Theorem 3.1, it is clear that whenever the environment and the policy are smooth, then the value functions are also smooth. A natural and important follow-up question to ask is whether this claim holds in the reverse direction. More concretely, we ask whether it is reasonable to seek a policy that is also smooth with respect to the state while maximizing the reward. If the claim holds true, then seeking a smooth policy can serve as an efficient and unbiased prior knowledge, that can help us reduce the policy search space significantly, while still guaranteeing that we are searching for high-performing policies.

• **Existence of smooth and nearly-optimal policies.** The following result shows that for any $\epsilon > 0$, there exists an $\epsilon$-optimal policy that is $\mathcal{O}(L_Q/\epsilon)$ smooth, where $L_Q$ defined in Theorem 3.1 only depends on the smoothness of reward and transition. This structural observation naturally suggests seeking a smooth policy for smooth environments.

**Theorem 3.2.** *Suppose the environment is $(L_r, L_\mathbb{P})$-smooth. Then for any $\epsilon > 0$, there exists an $\epsilon$-optimal policy $\pi$ that is also smooth, i.e.,*

$$V^*(s) - V^\pi(s) \le \frac{2\epsilon}{1-\gamma}, \ \forall s \in \mathcal{S} \quad \text{and} \quad \|\pi(\cdot|s) - \pi(\cdot|s')\|_1 \le |\mathcal{A}| \log|\mathcal{A}| L_Q \|s - s'\| /\epsilon,$$

*where $L_Q$ is defined as in Theorem 3.1.*

Notably, the proof of Theorem 3.2 relies on the key observation that any smooth $Q$-function satisfying (1) can be fed into the softmax operator, which induces a smooth policy. This observation also provides a way for value-based methods (e.g., Q-learning) to learn a smooth policy. Namely, one can first learn a smooth surrogate of the optimal $Q$-function, and then feed the learned surrogate into the softmax operator to induce a close-to-optimal policy that is also smooth.

---

* See Remark 3.2 for discussions when the smoothness property holds approximately.

- **Robustness against observation noise with smooth policies.** We have so far established that smooth policies naturally exist in a smooth environment as close-to-optimal policies, and thus smoothness serves as a strong prior for policy search. We will further demonstrate that the benefits of smooth policy go beyond boosting learning efficiency, by bringing in the additional advantage of robustness against observation uncertainty.

**Theorem 3.3.** *Let $\pi(a|s)$ be $L_\pi$-smooth policy. For any perturbation sequence $\{\delta_s^t\}_{t \geq 0, s \in \mathcal{S}}$, define a perturbed policy (non-stationary) $\widetilde{\pi} = \{\widetilde{\pi}_t\}_{t \geq 0}$ by*

$$\widetilde{\pi}_t(a|s) = \pi(a|s + \delta_s^t),$$

*with $\|\delta_s^t\| \leq \epsilon$ for all $t \geq 0$. Accordingly, define the value function of the non-stationary policy $\widetilde{\pi}$*

$$V^{\widetilde{\pi}}(s) = \mathbb{E}\big[ \sum_{t=0}^\infty \gamma^t r(s_t, a_t) | s_0 = s, a_t \sim \widetilde{\pi}_t(\cdot|s_t + \delta_{s_t}^t),$$
$$s_{t+1} \sim \mathbb{P}(\cdot|s_t, a_t)\big].$$

*Then we have $|V^\pi(s) - V^{\widetilde{\pi}}(s)| \leq \frac{2L_\pi \epsilon}{(1-\gamma)^2}$, Similarly, we have*

$$|Q^\pi(s, a) - Q^{\widetilde{\pi}}(s, a)| \leq \frac{2L_\pi \epsilon}{(1-\gamma)^2},$$

*where $Q^{\widetilde{\pi}}$ is defined similarly as $V^{\widetilde{\pi}}$.*

Theorem 3.3 establishes the following fact: for a discounted MDP with finite state and finite action space, the value of the policy when providing the perturbed state is close to the value of the policy when given the non-perturbed state, provided the policy is Lipschitz continuous in its state. As an important implication, the learned smooth policy will be robust in the state observation, in the sense that the accumulated reward will not deteriorate much when noisy, or even adversarially constructed state observations are given to the policy upon decision making.

We emphasize that Theorem 3.3 holds without any smoothness assumption on the transition or the reward function. It should also be noted that there are various notions of robustness in MDP, e.g., robustness against changes in transition kernel [34; 17], which we defer as future investigations.

Before we conclude this section, we briefly remark on certain generality of our discussion.

*Remark* 3.1 (**Applicability to MARL**). The discussions in this section do not depend on the size of the state space, and apply to the multi-agent setting without any change. To see this, note that our discussion holds for any discrete state-action space. Setting $\mathcal{S}$ as the joint state space and $\mathcal{A}$ as the joint action space, then the obtained results trivially carry over to the cooperative MARL setting.

*Remark* 3.2 (**Environments with approximate smoothness**). Many environments are partially smooth, in the sense that the transition or the reward is non-smooth only on a small fraction of the state space. Typical examples include the Box2D environment [33], where the agent receives smooth reward when in non-terminal states (airborne for Lunar Lander), and receives a lump-sum reward in the terminal state (land/crash) – a vanishing fraction of the entire state space. Given the environment being largely smooth, it should be expected that for most states the optimal policy is locally smooth. Consequently, inducing a smoothness prior serves as a natural regularization to constrain the search space when solving these environments, without incurring a large bias.

*Remark* 3.3 (**Non-smooth environments**). From the perspective of robust statistics, achieving robustness often necessitates a certain level of smoothness in the learned policy, regardless of the smoothness of the optimal policy. In scenarios where the environment itself is non-smooth, the optimal policy can also be non-smooth. However, it is important to note that such non-smooth optimal policies are typically not robust. This means that by trading-off between the approximation bias and robustness, the smooth policy learnt by out method has the potential to outperform non-smooth policies in perturbed environments.

## 4  Method

In this section, we propose our robust MARL framework, advErsarially Regularized multiageNt reInforcement lEarning (ERNIE).

### 4.1  Learning Robust Policy with ERNIE

Section 3 shows that the robustness of a policy depends on its Lipschitz constant. Therefore, in ERNIE we propose to control the Lipschitz constant of each policy with adversarial regularization.

Given a policy $\pi_{\theta_k}$, where $k$ is the agent index, the ERNIE regularizer is defined by

$$R_\pi(o_k; \theta_k) = \max_{||\delta|| \leq \epsilon} D(\pi_{\theta_k}(o_k + \delta), \pi_{\theta_k}(o_k)). \tag{2}$$

Here $\delta(o_k, \theta_k)$ is a perturbation adversarially chosen to maximize the difference between the policy's output for the perturbed observation $o_k + \delta(o_k, \theta_k)$ and the original observation $o_k$. In this case $\epsilon$ controls the perturbation strength and $|| \cdot ||$ is usually taken to be the $\ell_2$ or $\ell_\infty$ norm. Note that $R_\pi(o_k; \theta_k)$ essentially measures the local Lipschitz smoothness of policy function $\pi_\theta$ around the observation $o_k$, defined in metric $D(\cdot, \cdot)$. Therefore minimizing $R_\pi(o_k; \theta_k)$ will encourage the policy to be smooth.

Regularization (2) allows straightforward incorporation into MARL algorithms that directly perform policy search. For actor-critic based policy gradient methods, the regularizer (2) can be directly included into the objective for updating the actor (policy) networks. When optimizing stochastic policies (e.g., MAPPO [35]), $D$ can be taken to be the KL divergence and for deterministic policies (e.g., MADDPG [36] or Q-learning [37]), we set $D$ to be the $\ell_p$ norm.

More concretely, let $\mathcal{L}(\theta)$ denote the policy optimization objective, i.e., the negative weighted value function of the policy. We then augment $\mathcal{L}(\theta)$ with (2), and minimize the regularized objective

$$\min_\theta \mathcal{F}(\theta) = \mathcal{L}(\theta) + \lambda \sum_{n=1}^{N} \mathbb{E}_{\pi_n}\left[R_\pi(o_n; \theta_n)\right], \tag{3}$$

where $\lambda$ is a hyperparameter. We remark that Shen et al. [3] has explored similar regularization for single-agent RL (with a goal of improving sample efficiency), but as we explain in sections 4.2, 4.3, and 4.4, successful application to MARL robustness is highly non-trivial.

### 4.2 Stackelbeg Training with Differentiable Adversary

Although accurately solving (3) will result in a high-performing and robust policy, we note that (3) is a nonconvex-nonconcave minimax problem. In practice, we can use multiple steps of projected gradient ascent to approximate the worse-case state perturbation $\delta(o_k, \theta_k)$, followed by one-step gradient descent for updating the policies/Q-function. Even though this optimization method already significantly improves robustness over the baseline algorithms, we found that the training process could be quite unstable. We hypothesize that the intrinsic instability of MARL algorithms due to simultaneous updates of multiple agents is greatly amplified by the non-smooth landscape of adversarial regularization.

To promote a more stable straining process, we propose to reformulate adversarial training in ERNIE as a Stackelberg game. The reformulation defines adversarial regularization as a leader-follower game [38]:

$$R_\pi(o, \delta_\theta^K(o); \theta) = D\big(\pi_\theta(o + \delta^K(o, \theta)), \pi_\theta(o)\big) \tag{4}$$
$$\text{s.t.} \quad \delta^K(o, \theta) = \underbrace{U_\theta \circ U_\theta \circ \cdots \circ U_\theta}_{\text{K-fold composition}}(\delta^0(o, \theta)).$$

Here $\circ$ denotes the operator composition (i.e $f \circ g = f(g(\cdot))$), and

$$\delta^{k+1}(o, \theta) = U_\theta(\delta^k(o, \theta)) = \delta^k(o, \theta) + \eta \nabla_\delta D\left(\pi_\theta\left(o + \delta^k(o, \theta)\right), \pi_\theta(o)\right)$$

is a one-step gradient ascent for maximizing the divergence of the perturbed and original observation.

Compared to the vanilla adversarial regularizer in (2), the perturbation $\delta$ is treated as a function of the model parameter $\theta$. This formulation allows the leader ($\theta$) to anticipate the action of the follower ($\delta$), since the follower's response given observation $o$ is fully specified by $\delta^K(o, \theta)$. This structural anticipation effectively produces an easier and smoother optimization problem for the leader ($\theta$), whose gradient, termed Stackelberg gradient, can be readily computed by

$$\frac{\partial R_\pi(o, \delta_\theta^K(o); \theta)}{\partial \theta} = \underbrace{\frac{\partial R_\pi(o, \delta^K, \theta)}{\partial \theta}}_{\text{leader}} + \underbrace{\frac{\partial R_\pi(o, \delta^K(\theta), \theta)}{\partial \delta^K(\theta)} \frac{\delta^K(\theta)}{\partial \theta}}_{\text{leader-follower interaction}}$$

Note that the gradient used in (3) only contains the "leader" term, such that interaction between the model $\theta$ and the perturbation $\delta$ is ignored. The computation of the Stackelberg gradient can be reduced to Hessian vector multiplication using finite difference method [39], which only requires two

backpropogations and extra $\mathcal{O}(d)$ complexity operation. Thus no significant computational overhead is introduced for solving (4).

The benefit of Stackelberg training for MARL is twofold. First, a smoother optimization problem results in a more stable training process. This extra stability is essential given the inherent instability of MARL training. Second, giving the policy $\theta$ priority over the attack $\delta$ during the training process allows for a better training data fit than normal adversarial training allows. This better fit allows the MARL policies trained with Stackelberg training to perform better in lightly perturbed environments than those trained with normal adversarial regularization.

### 4.3 Robustness against Malicious Actions

Given the complex interactions of agents within of a multi-agent system, a robust policy for any given agent should meet the criterion that the action made is not overly dependent on any small subset of agents. This is particularly the case when the agents are homogeneous in nature [4; 40], and thus there should be no notion of *coreset agents* in the decision-making process that could heavily influence the actions of other agents. We proceed to show how ERNIE could be adopted to induce such a notion of robustness.

The core idea of ERNIE for this scenario is to encourage policy/Q-function smoothness with respect to *joint actions*. Similar to our treatment in (2), we now seek to promote learning a Q-function that yields a consistent value when perturbing the actions for any small subset of agents. Specifically, for discrete action space, we define a regularizer on the global Q-function as

$$R_\omega^A(s, \mathbf{a}) = \max_{D(\mathbf{a}, \mathbf{a}') \leq K} ||Q(s, \mathbf{a}; \omega) - Q(s, \mathbf{a}'; \omega)||_2^2, \tag{5}$$

where $D(\mathbf{a}, \mathbf{a}') = \sum_i I(\mathbf{a}_i \neq \mathbf{a}'_i)$. The regularizer (5) seeks to compute the worst subset of changed actions with cardinality less than $K$. For continuous action spaces, one could replace the metric $D$ in (5) by a differentiable metric defined over the action space (e.g., $||\cdot||_2$-norm), and then evaluate the regularizer with projected gradient ascent.

To evaluate the adversarial regularizer for the discrete action space, we propose to solve (5) in a greedy manner by finding the worst-case change in the action of a single agent at a time, until the action of $K$ agents is changed. Specifically, at each training step, we search through all the agents/actions and then pick the actions that produce the top-$K$ changes in the Q-function, resulting in a $\mathcal{O}(|\mathcal{A}| * N * K)$ computation. Our complete algorithm can be found in Appendix H, and we find that in our numerical study, perturbing the action of a single agent ($K = 1$) is sufficient for increased robustness.

Similar to the regularizer in (2), the regularizer in (5) provides the benefits of Lipschitz smoothness (with respect to the Hamming distance) and data augmentation with adversarial examples. If the behavior of a few agents changes (either maliciously or randomly), the behavior of policies trained by conventional methods may change drastically. On the other hand, policies trained by our method will continue to make reasonable decisions, resulting in more stable performance (see section 5).

### 4.4 Extension to Mean-field MARL

MARL algorithms have been known to suffer from the curse of many agents [4], as the search space of policies and value functions grows exponentially w.r.t. the number of agents. A practical approach to tackle this challenge of scale is to adopt the mean-field approximation, which views each agent as realizations from a distribution of agents. This distributional perspective requires a distinct treatment of ERNIE applied to the mean-field setting.

Mean-field MARL avoids the curse of many agents by approximating the interaction between each agent and the global population of agents with that of an agent and the average agent from the population. In particular, we can approximate the action-value function of agent $j$ as $Q^j(\mathbf{s}, \mathbf{a}) = Q^j(s_j, d_s, a_j, \bar{a}_j)$, where $\mathbf{a}$ is the global joint action, $\mathbf{s}$ is the global state, $\bar{a}_j$ is the average action of agent $j$'s neighbors, and $d_s$ is the empirical distribution of states over the population. Such an approximation has found widespread applications in practical MARL algorithms [41; 42; 40; 43], and can be motivated in a principled fashion for agents of homogeneous nature [4; 40].

To learn robust and scalable policies, we extend ERNIE to the mean-field setting by applying adversarial regularization to the approximation terms $d_s$ and $\bar{a}^j$. It is important to note that as the terms $d_s$ and $d'_s$ represent distributions over states, we bound the attack by the Wasserstein distance [44]. In what follows we only apply the regularizer to $d_s$ for simplicity. This leads to a new regularizer

defined over the mean field state

$$R_{\mathcal{W}}^Q(s;\theta) = \max_{\mathcal{W}(d'_s,d_s)\leq\epsilon} \sum_{a\in\mathcal{A}} \|Q_\theta(s,d'_s,a) - Q_\theta(s,d_s,a)\|_2^2,$$

where $\mathcal{W}$ denotes the Wasserstein distance metric. Since the explicit Wasserstein constraint may be difficult to optimize in practice, we can instead enforce the constraint through regularization, as displayed in Appendix C.

## 5 Experiments

We conduct extensive experiments to demonstrate the effectiveness of our proposed framework. In each environment, we evaluate MARL algorithms trained by the ERNIE framework against baseline robust MARL algorithms. To evaluate robustness, we train MARL policies in a non-perturbed environment and evaluate these policies in a perturbed environment. The reported results are gathered over five runs for each algorithm. Given the space limit, we put additional results in Appendix E.

**Traffic Light Control.** In this scenario, cooperative agents learn to minimize the total travel time of cars in a traffic network. We use QCOMBO [45] and COMA [46] (results in appendix E) as our baseline algorithms and conduct experiments using the Flow framework [47]. We train the MARL policies on a two-by-two grid (four agents). We then evaluate the policies on a variety of realistic environment changes, including different car speeds, traffic flows, network topologies, and observation noise. In each setting, we plot the reward for policies trained with ERNIE, the baseline algorithm (QCOMBO), and another baseline where the attack $\delta$ is generated by a Gaussian random variable (Baseline-Gaussian, see Appendix I). Implementation details can be found in Appendix G.

Figure 1, 2a, and 2b show that the baseline algorithm is vulnerable to small changes in the training environment (higher reward is better). On the other hand, ERNIE achieves more stable reward on each environment change. This observation confirms that the ERNIE framework can improve robustness against observation noise and changing transition dynamics. The Gaussian baseline performs well on some environment changes, like when the observations are perturbed by Gaussian noise, but performs poorly on other environment changes, like when the car speed is changed. We hypothesize that while some environment changes may be covered by Gaussian perturbations, other environment changes are unlike Gaussian perturbations, resulting in a poor performance from this baseline.

**Robustness Against Malicious Actions.** We also evaluate the extension of ERNIE to robustness against changing agent behavior, which we refer to as ERNIE-A. To change agent behavior, we adversarially change the action of a randomly selected agent a small percentage of the time, i.e. 5% or 3% of the time. As can be seen in Figures 2c and 2d, the two baseline algorithms perform poorly when some agent's behavior changes. In contrast, ERNIE-A is able to maintain a higher reward.

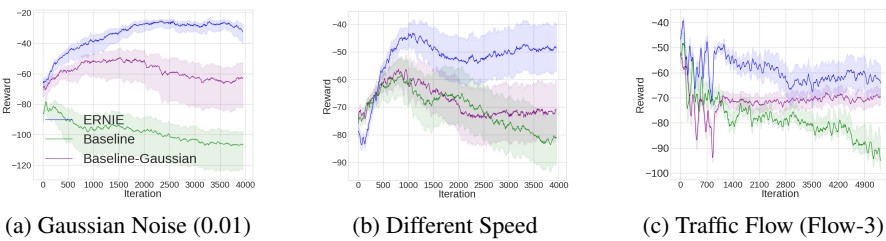

(a) Gaussian Noise (0.01)      (b) Different Speed      (c) Traffic Flow (Flow-3)

Figure 1: Evaluation curves on different environment changes for traffic light control.

**Particle Environments.** We evaluate ERNIE on the cooperative navigation, predator-prey, tag, and cooperative communication tasks. In each setting, we investigate the performance of the baseline algorithm (MADDPG), ERNIE, M3DDPG, and the baseline-gaussian in environments with varying levels of observation noise. We also compare ERNIE to the RMA3C algorithm proposed in [23]. In Figure 3, we find ERNIE performs better or equivalently than MADDPG in all settings. Surprisingly, M3DDPG can provide some robustness against observation noise, even though it is designed to provide robustness against malicious actions.

**Mean-field MARL.** We evaluate the performance of the mean-field ERNIE extension on the cooperative navigation task [36] with different numbers of agents. We compare the performance of the baseline algorithm, ERNIE, and M3DDPG under various levels of observation noise. We use

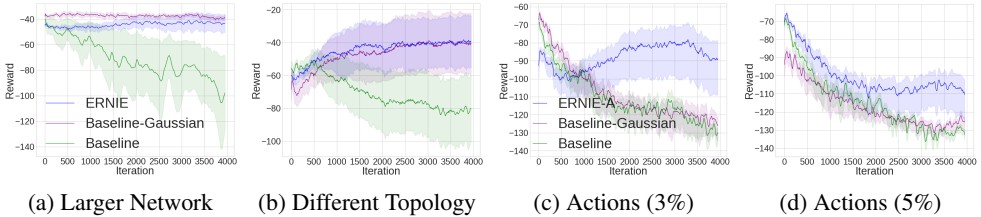

(a) Larger Network     (b) Different Topology     (c) Actions (3%)     (d) Actions (5%)

Figure 2: Performance on changed traffic network topologies and with malicious agents. In Figures (c) and (d) we perturb the actions according to the specified percentages.

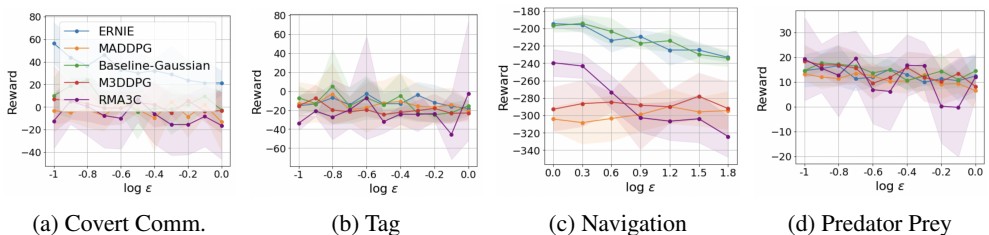

(a) Covert Comm.     (b) Tag     (c) Navigation     (d) Predator Prey

Figure 3: Training reward versus noise level ($\epsilon$) in the evaluation environment for the particle games.

mean-field MADDPG as our baseline and follow the implementation of [40]. As can be seen in Figure 4, ERNIE displays a higher reward and a slower decrease in performance across noise levels.

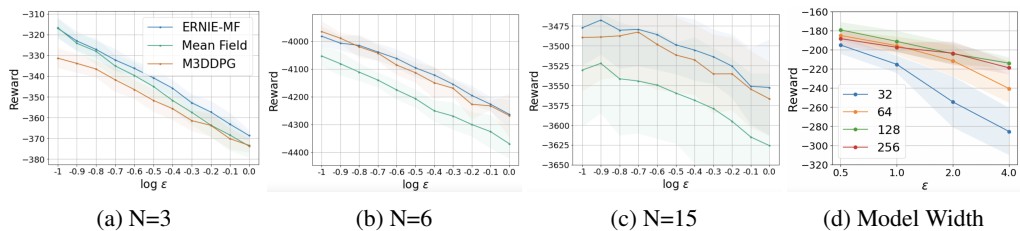

(a) N=3     (b) N=6     (c) N=15     (d) Model Width

Figure 4: (a)-(c) Training reward versus noise level (mean $\pm$ standard deviation over 5 runs) with a various number of agents (N) (d) Network width and robustness.

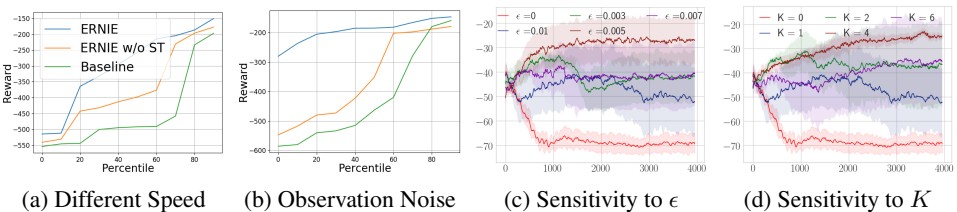

(a) Different Speed     (b) Observation Noise     (c) Sensitivity to $\epsilon$     (d) Sensitivity to $K$

Figure 5: Sensitivity and ablation experiments.

**Hyperparameter Study.** We investigate the sensitivity of ERNIE to the hyperparameters $K$ (the number of attack steps) and $\epsilon$ (the perturbation strength). We plot the performance of different hyperparameter settings in the traffic light control task with perturbed car speeds on three random seeds. From Figures 5c and 5d we can see that adversarial training ($K > 0$) outperforms the baseline ($K = 0$) for all $K$. Similarly, we can see that all values of $\epsilon$ outperform the baseline ($\epsilon = 0$). This indicates that ERNIE is robust to different hyperparameter settings of $\epsilon$ and $K$.

**Sensitivity and Ablation Study.** The advantage of the ERNIE framework goes beyond improving the mean reward. To show this, we evaluate 10 different initializations of each algorithm in two traffic environments: one with different speeds and another environment with observation noise. We then sort the cumulative rewards of the learned policies and plot the percentiles in Figures 5a and 5b. Although the best-case performance is the same for all algorithms, ERNIE significantly improves the

robustness to failure. As an ablation, we evaluate the effectiveness of ERNIE with and without the Stackelberg formulation of adversarial regularization (ST). ERNIE displays better performance in both settings, indicating that Stackelberg training can lead to a more stable training process. Ablation experiments in other environments can be found in Appendix E.4.

**Robustness and Network Width.** In Appendix B, we show that in order to learn a robust policy with ERNIE, we should use a sufficiently wide neural network. Therefore, we evaluate the robust performance of ERNIE using policy networks with 32, 64, 128, and 256 hidden units. We carefully tune their regularization parameters such that all networks perform similarly in the lightly perturbed environment. As seen in Figure 4d, when the perturbed testing environment deviates more from the training environment, the performance of the narrower policy networks (32/64 hidden units) significantly drops, while the wider networks (128/256 hidden units) are more stable. We also observe that when the policy networks are sufficiently wide (128/256), their robustness is similar.

**Robotics Experiments.** Additional experiments in multi-agent drone control environments can be found in Appendix E.1, which further verify the enhanced robustness that ERNIE provides.

## 6 Discussion

ERNIE is motivated by smoothness, but real-world environments are not always smooth. In section 3, we hypothesize that most environments are at least partially smooth, implying that smoothness can serve as useful prior knowledge while providing robustness (our experiments validate this). To increase ERNIE's flexibility, future work could adaptively select $\lambda$ based on the current state to allow for state-dependent smoothness.

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
