# OpenReview forum: "Robust Multi-Agent Reinforcement Learning via Adversarial Regularization: Theoretical Foundation and Stable Algorithms"
_NeurIPS.cc/2023/Conference — NeurIPS 2023 poster_

### Official Review · Reviewer_YAu4 · 2023-06-22

**Soundness:** 3 good
**Presentation:** 3 good
**Contribution:** 3 good
**Rating:** 6
**Confidence:** 2

**Summary:**

The paper proposes a training objective Lipschitz regularization for obtaining adversarial robustness in cooperative multi-agent RL. To overcome training instablities, the paper reframes the regularization as a Stackleberg game. The paper also shows how the method can be applied to Mean-field approximations of some MARL training algorithms and carries out two experiments on the traffic and the multi-agent Particle Environments.

**Strengths:**

- Nice background on connecting Lipschitz continuity with robustness with respect to different variables of game and policy.
- Overall solid method aiming to overcome the instability of adversarial training in the context of MARL.

**Weaknesses:**

I believe the clarity of the experiments section could be significantly improved. The figures are barely legible due to the small (font) size. Moreover, the figure caption could be improved, e.g., does "Evaluation curves" refer to "Learning curves" as in the episode reward over the training? Moreover, what does "Baseline-Gaussian" refer to?

**Questions:**

The method seems to be focused on Cooperative Markov Games. Can you elaborate on whether, how, or why the method would apply to other multi-agent games (competitive or other objectives, etc.)?

How does ERNIE compare to direct applications on Eq (2) and (3) in the training process?

**Limitations:**

Readability of the figures could be improved

---

> ### Author Rebuttal · Authors · 2023-08-10
>
> Dear Reviewer YAu4,
>
> - **Weakness 1: Clarity of figures.**
> Thank you for pointing this out! We will increase the font size on the figures to improve the legibility. As for "Evaluation Curves," this refers to the reward achieved when we evaluate the policy in the perturbed environment. We also just realized "training reward" in the captions of Figures 3 and 4 should be replaced with "evaluation reward." Thank you for catching this! Finally, Baseline-Gaussian refers to a Baseline where the perturbation in ERNIE is sampled from a Gaussian distribution.
>
> - **Question 1: Could the method be applied to competitive multi-agent games?**
> Yes, ERNIE can straightforwardly be applied to non-cooperative environments. However, the theoretical analysis may become significantly more complicated in such settings. Therefore we leave applications to non-cooperative settings for future work.
>
> - **Question 2: How does ERNIE compare to direct application of equations (2) or (3)?**
> Please note that we evaluate the direct application of (2) and (3) into training in line 358-374 and Figures 5a-b (this baseline is denoted as ERNIE w/o ST). As we can see, ERNIE significantly outperforms this baseline, showing the benefit of our proposed Stackelberg training in ERNIE.

---

### Official Review · Reviewer_fpJY · 2023-07-05

**Soundness:** 3 good
**Presentation:** 2 fair
**Contribution:** 2 fair
**Rating:** 5
**Confidence:** 4

**Summary:**

The paper introduces a robust Multi-Agent Reinforcement Learning (MARL) framework called ERNIE (adversarially Regularized multi-ageNt reInforcement lEarning). The goal of ERNIE is to address the issue of policy robustness in MARL algorithms. The key contributions of the paper are as follows:

1. Robust MARL Framework: The authors propose ERNIE, which promotes the Lipschitz continuity of policies through adversarial regularization. This regularization enhances robustness against noisy observations, changing transition dynamics, and malicious actions of agents.
2. Lipschitz Control: ERNIE controls the Lipschitz constant of policies, allowing them to be more resilient to variations in the environment. This control is achieved by leveraging adversarial regularization, which encourages Lipschitz continuity with respect to state observations and actions.
3. Training Stability: The paper addresses the training instability that adversarial regularization can introduce by reformulating it as a Stackelberg game. This reformulation provides a more stable training process, making the optimization problem smoother for the defender (policy).
4. Experimental Evaluation: The authors extensively evaluate the effectiveness of ERNIE in traffic light control and particle environments. The experiments demonstrate that ERNIE improves robustness and performance compared to non-robust MARL approaches.
5. Extension to Mean-field MARL: ERNIE is extended to mean-field MARL, a strategy to scale up MARL algorithms.
6. Theoretical Insights: The paper provides theoretical insights into the smoothness and robustness of policies in MARL. It shows that in environments with smooth transition and reward functions, policies' value functions are also smooth. The authors prove the existence of smooth and close-to-optimal policies in such environments and establish that policy robustness is proportional to its Lipschitz constant.

**Strengths:**

**Originality**: The theoretical insights into the smoothness of environments and robustness of policies look interesting to me. By assuming the smoothness of environments, we can get some good properties of robustness. This idea is very interesting to me. However, this idea is very similar to the paper [1]“****Deep Reinforcement Learning with Robust and Smooth Policy****”.

[1] Shen, Q., Li, Y., Jiang, H., Wang, Z., & Zhao, T. (2020, November). Deep reinforcement learning with robust and smooth policy. In *International Conference on Machine Learning* (pp. 8707-8718). PMLR.

**Quality**: The paper is very informative. I can see the authors have a lot to show readers in the paper: the theoretical results, the proposed framework, how to solve the problem efficiently, and how to adopt the proposed framework to large-scale multi-agent systems.

**Clarity**: The paper appears to be clear in its presentation of the problem, the proposed solution, and the results of the experiments. The authors provide a theoretical motivation for their algorithm, explaining how Lipschitzness (smoothness) serves as a natural way to gain robustness while reducing the policy search space. The extension of ERNIE to mean-field MARL is also clearly explained.

**Significance**: I believe this paper's theoretical analysis of how smoothness helps in training robust policies would help future researchers in the field of robust MARL to develop better solutions to the robustness issues.

**Weaknesses:**

**Confusing Writing**: I found the paper sometimes inconsistent in motivation, methods, and experiments, which makes me confused about what problem this paper tries to solve. In the abstract, the authors point out that we need to consider real-to-sim gaps in the deployment of MARL algorithms, I quote the original sentence there “This presents a serious concern for the real world deployment of MARL algorithms, where the testing environment may slightly differ from the training environment.” But later, the authors claim that their proposed method is robust to perturbations on observations and actions. I also quote the original sentences as follows: “Based on these insights, we propose a new robust MARL framework, ERNIE, that promotes the Lipschitz continuity of the policies with respect to the state observations and actions by adversarial regularization.” and “In particular, we develop an adversarial regularizer to minimize the discrepancy between each policy’s output given a perturbed observation and a non- perturbed observation.” When I dive into the methodology of this paper, I found most of the contexts are about **Robustness against observation noise with smooth policies** and **Robustness against Malicious Actions.** We can see that the abstract and some texts in the introduction mislead readers about what kind of robustness this paper tries to achieve in MARL.

**Evaluations on Simple Games**: The evaluation of the proposed methods is conducted on specific tasks such as traffic light control and particle environment tasks. While these tasks are complex, they may not fully reveal insights into how the smoothness of environments affects the robustness of policies.  I would suggest running experiments on some simple matrix games or small-scale multi-agent toy examples [1] to better study how smoothness is connected to robustness.

[1] Hu J, Wellman MP. Nash Q-learning for general-sum stochastic games. Journal of machine learning research. 2003;4(Nov):1039-69.

**Lack of Comparison with Robust MARL Methods:** This paper claims that the proposed method is robust to perturbed actions and observations. Considering there are some existing works about robust MARL considering adversarial-purebred observations [1, 2, 3] and perturbed actions [4], should the authors compare the proposed method with at least one of these robust MARL methods?

[1] Han S, Su S, He S, Han S, Yang H, Miao F. What is the solution for state adversarial multi-agent reinforcement learning?. arXiv preprint arXiv:2212.02705. 2022 Dec 6.

[2] He S, Han S, Su S, Han S, Zou S, Miao F. Robust Multi-Agent Reinforcement Learning with State Uncertainty. Transactions on Machine Learning Research. 2023 Mar 10.

[3] Zhou Z, Liu G, Zhou M. A Robust Mean-Field Actor-Critic Reinforcement Learning Against Adversarial Perturbations on Agent States. IEEE Transactions on Neural Networks and Learning Systems. 2023 Jun 5.

[4] Li, S., Guo, J., Xiu, J., Yu, X., Wang, J., Liu, A., ... & Liu, X. (2023). Byzantine Robust Cooperative Multi-Agent Reinforcement Learning as a Bayesian Game. *arXiv preprint arXiv:2305.12872*.

**Writing Suggestions:**

The mean-field MARL part does not tightly connect to the main text of this paper. Would it be better to make it a remark instead of a subsection? Besides, the organization of this paper makes me confused, for example, Robustness against observation noise with smooth policies (a paragraph) and Robustness against Malicious Actions (a subsection) are placed on different layers of this paper. Should they be on similar layers? As a paper studying robust MARL, in the related work, this paper uses more text to discuss robust RL works than robust MARL. The literature review of robust MARL is not comprehensive.

**Questions:**

It would be great if the authors can resolve my concerns in the weakness box and the Originality concerns in the strength box. I will decide my final score after the rebuttal phase according to the authors' responses.

**Limitations:**

NAN

---

> ### Author Rebuttal · Authors · 2023-08-10
>
> Dear Reviewer fpJY,
>
> Thank you for your very detailed and helpful review! We believe that by addressing your critiques, we have significantly improved the paper. In particular, we clarify the originality of our method, add comparisons with robust MARL methods, and provide ways to clear up the writing.
>
> - **Weakness 0: Difference from Shen et al., 2020 [1] (Clarification on Originality).**
> The work of [1] is one of the motivating works from which we base our work. However, we believe we have provided several significant contributions over [1]. We list them as follows: (1) [1] provides no theoretical results to support their work. In constrast, we develop theory as to why smoothness provides robustness and can boost sample efficiency. (2) [1] is developed for the single-agent case and does not consider the instability of MARL training. To improve upon this, we develop a Stackelberg extension to adversarial regularization which can significantly boost performance. (3) We develop extensions to mean-field MARL and action robustness, which [1] does not consider.
>
> - **Weakness 1: Confusing Writing.**
> We apologize for any confusion in the introduction. By bridging the sim-to-real gap, we meant to refer to the cases where the simulator does not contain any observation noise but the real world environment does (or contains noise in the actions). We remark that this is a realistic part of the sim-to-real gap (i.e. cameras in simulation may not pick up any dust particle, but in the real world they may). To avoid confusion, we will state precisely what type of robustness we are attempting to achieve, including the specific sim-to-real robustness we seek.
>
> - **Weakness 2: Evaluations on Simple Games.**
> One of the main benefits of our algorithm over previous works such as [1] is that we improve the stability of the deep MARL training and develop adversarial regularization for MARL. In matrix environments, it may be very difficult to evaluate these gains as deep MARL is not needed. However, we do conduct experiments on the simple spread environment, which has a very simple objective (for the agents to cover landmarks). In this environment (Figure 3c), we can observe that smoothness not only improves robustness, but also clean performance.
>
> - **Weakness 3: Lack of Comparison with Robust MARL Methods.**
> This is a great suggestion. We apply the method RMA3C from [1] to our particle environment experiments. We remark that this algorithm is quite similar to the RMAAC algorithm in [2], which we discuss in response to question 2 of J2jx. The experimental results of RMA3C can be seen in Figure 2 of the rebuttal material. We notice that RMA3C can provide some robustness, but cannot outperform ERNIE. We hypothesize this is because RMA3C trains on perturbed observations which may result in underfitting. On the other hand, ERNIE trains with clean observations and applies regularizations to the perturbed observations, allowing better fitting to the non-perturbed environment while still providing robustness. With the addition of this baseline, we have compared our algorithm with three robust MARL baselines.
>
> - **Weakness 4: Writing Suggestions.**
> We apologize for the confusion. These suggestions make sense to us. We can make the mean-field MARL section a remark, as no sections build upon it, and it is more of an extension of ERNIE to a special case. In addition, we agree that robustness against observation noise can be put in similar layers of the paper (subsections). Finally, we believe that by adding the discussion on the related literature you suggested in weakness 3 ([1-4]), our robust MARL literature review will be more comprehensive.
>
> Please let us know if we can add further experiments or discussion to help resolve your concerns about our work.

---

> > ### Author Response · Authors · 2023-08-14
> > **Follow up on Rebuttal**
> >
> > Dear Reviewer fpJY,
> >
> > Thank you again for your insightful comments! We believe that by addressing your concerns about originality, the writing, simple environments, and comparisons with other robust MARL works, we have greatly strengthened our work. As the discussion period is almost halfway over, we kindly inquire if you have any follow-up questions or comments. If we have addressed your concerns, we would really appreciate it if you would consider raising your score.

---

> > > ### Comment · Reviewer_fpJY · 2023-08-15
> > > **Thank you for your response**
> > >
> > > I appreciate the authors' thorough response and the inclusion of additional experiments. With my concerns addressed, I'm now inclined to give a rating of 5 (the original rating is 4). I also kindly request the authors to incorporate the supplementary experiment results and other details into the camera-ready version.

---

> > > > ### Author Response · Authors · 2023-08-15
> > > > **Thank you for the discussion**
> > > >
> > > > Dear Reviewer fpJY,
> > > >
> > > > Thank you for the engaging discussion. We will be sure to incorporate the supplementary experimental results and other details into the camera-ready version.

---

### Official Review · Reviewer_J2jx · 2023-07-07

**Soundness:** 3 good
**Presentation:** 3 good
**Contribution:** 2 fair
**Rating:** 6
**Confidence:** 3

**Summary:**

Robust RL is attracting great attention, and lately there has been great advancements both in theory and empirically. However, there is only limited work in robust multi-agent reinforcement learning. Under some assumptions, the paper shows the existence of policies with nice properties. Building upon this, the authors propose a training framework that includes adversarial perturbation on observation in order to increase robustness of the learned policy.

**Strengths:**

The main ideas behind this formulation are quite intuitive. The paper also contains insightful theoretical component to support such intuition. The algorithm is shown to be promising in simulations.

**Weaknesses:**

See limitations.

**Questions:**

1. I find that calling lipschitzness smoothness quite confusing. In particular, in Definition 3.1, the "smoothness" is actually defined as the reward and dynamics being $L_r,L_{\mathbb{P}}$-Lipschitz. Subsequently, in the writing, there are many places where the authors interchange the smoothness and lipschitzness, e.g., line 187-188. As far as I know, more commonly smoothness is about the gradient of a function being Lipschitz. Is there a strong reason for this particular choice of words?
1. This recent work [1] considers a Markov game with state perturbation adversaries. How does it compare to your work?
[1] Sihong He, Songyang Han, Sanbao Su, Shuo Han, Shaofeng Zou, and Fei Miao. Robust multi-agent reinforcement learning with state uncertainty. Transactions on Machine Learning Research, 2023.
1. What will be the main difficulty in extending ERNIE to continuous state and action setting?
1. The main idea of making the learned policy Lipschitz is somewhat similar to the idea of smoothening the decision boundary. I would like to think this as the robustness in the "average" case rather than the worst case. Is there any work in the MARL literature that considers the worst case robustness?

**Limitations:**

1. In the paper survey section regarding the single-agent robust RL, I think the authors should include another important line of work: robust Markov decision process (RMDP). This distributionally robust reinforcement learning (DR-RL) framework provides a principled way to achieve robustness against perturbations in models. For example, [1,2] are considered to be the seminal theoretical works that established the framework. In this line of work, some works do not require expert knowledge of the underlying simulator. [3] is an offline DR-RL work that can even handle continous state and action empirically. [4,5] are policy gradient methods for DR-RL, and they also do not need prior expert knowledge.  On line 191-193, two DR-RL papers are included as reference but I think the authors should include some in the paper survey section as well.
[1] Garud N Iyengar. Robust dynamic programming. Mathematics of Operations Research, 30(2):257–280, 2005.
[2] Arnab Nilim and Laurent El Ghaoui. Robust control of Markov decision processes with uncertain transition matrices. Operations Research, 53(5):780–798, 2005.
[3] Panaganti, K., Xu, Z., Kalathil, D., & Ghavamzadeh, M. (2022). Robust reinforcement learning using offline data. Advances in neural information processing systems, 35, 32211-32224.
[4] Wang, Yue, and Shaofeng Zou. "Policy gradient method for robust reinforcement learning." International Conference on Machine Learning. PMLR, 2022.
[5] Navdeep Kumar, Esther Derman, Matthieu Geist, Kfir Levy, and Shie Mannor. Policy gradient for s-rectangular robust Markov decision processes. ArXiv preprint, abs/2301.13589, 2023. URL https://arxiv.org/abs/2301.13589.
1. Although the authors hint that the algorithm can tackle perturbations in dynamics/transition models, this is mainly an empirical claim with no theory to corroborate.
1. Lipschitz assumptions on reward and model are quite strong in my opinion. This is very unlikely to be true or verifiable in larger systems. Hence, the generalizability of the proposed method is limited.

---

> ### Author Rebuttal · Authors · 2023-08-10
>
> Dear Reviewer J2jx,
>
> Thank you for your very insightful review! We provide our responses below. We believe that by including discussions on the papers and works you mentioned, the quality of our paper is greatly improved.
>
> - **Question 1: Smoothness terminology.** Indeed, the smoothness in the optimization literature refers to the gradient being Lipschitz continuous, but this convention is perhaps made mainly within the optimization community. Here we choose the word smoothness as it conveys the intuition that the action made by the policy changes gradually as we vary the state. Kindly note that we do state the precise meaning in Definition 3.1.
>
> - **Question 2: How does [1] compare to our work?**
> Thank you for bringing this work to our attention, we were previously unaware of it. Our work and that of [1] are quite different and focus on different aspects of the robust MARL problem. Theoretically, [1] focuses on defining the robust equilibrium, and then showing that their proposed algorithm can converge. On the other hand, our work focuses on examining how smoothness affects robustness, and when smoothness can be used to boost the policies’ robustness. On the algorithmic side, [1] proposes to learn a separate adversarial policy for each agent, which learns to perturb that agent’s observation. This is quite different from our algorithm, which employs a Stackelberg version of adversarial regularization. Rather than learning a separate adversarial agent to perturb each policy as in [1], we can simply take a few steps of gradient ascent for each agent. We also remark that the algorithm developed in [1] does not put as large an emphasis on addressing the instability of robust MARL training as our work does. In addition, we provide extensions to mean-field MARL and action robustness, which [1] does not.
>
> - **Question 3: Extending ERNIE to continuous state and action.**
> Algorithmically, there is no difficulty in applying ERNIE to continuous state and action space. More specifically, ERNIE is already applied with continuous state spaces, and the regularizer defined in equation (2) can stay the same for continuous action spaces as well. Theoretically, it may be difficult to prove Theorem 3.2 in the case of continuous action spaces, as our current proof relies on feeding the optimal Q-function through the softmax operator.
>
> - **Question 4: Smoothing the decision boundary.** We agree with the smooth policy being related to smooth decision boundary, as a smooth decision boundary is typically a result of smooth decision function. To the best of our understanding there is no prior discussion in MARL literature that considers robustness in the worst-case setting, but we are happy to discuss this issue further if you know of any such works.
>
> - **Limitation 1:** Thank you for these suggestions! We agree that the provided works can help contextualize our work further. We will add discussion of the works [1-5] that you suggested.
>
> - **Limitation 2:** Indeed, for now this claim is mostly empirical and we do not have theory to back it up. However, we remark that this is quite an interesting and important finding, since it potentially indicates a cheap way to provide robustness against changing transition dynamics (which is a very difficult problem). In future work, we would like to develop theory regarding this empirical phenomena we observe. For now, we believe ERNIE provides robustness against transition perturbations due to the data-augmentation that happens when we add adversarially perturbed examples to the training set. Specifically, new scenarios emerge when the environment changes, and data augmentation with adversarial examples provides a large coverage of these scenarios as long as the environment change is small. Adapting to adversarial examples during training ensures that the agents will perform reasonably even in the worst case. Such reasoning can be found in line 61.
>
>
> - **Limitation 3:** Indeed, the smoothness assumption is unlikely to be completely satisfied in large-scale, complex MARL environments. However, we remark that smoothness is often a good approximation. This can be observed by the good performance of ERNIE in the traffic light control setting (a complex environment), where the reward is not smooth (it is the composition of several categorical functions) yet ERNIE still performs quite well.

---

> > ### Author Response · Authors · 2023-08-15
> > **Follow up on Rebuttal**
> >
> > Dear Reviewer J2jx,
> >
> > Thank you again for your insightful comments! We believe that by addressing your concerns on related works (in particular RMDP), clarifying our writing, and discussing some issues with extending ERNIE to continuous actions, our paper is stronger. We would also like to note that we have added a baseline algorithm similar to [1] in our response to Reviewer fpJY's review. We find that ERNIE is either competitive with this baseline or can outperform it in the particle environments.
> >
> > As the discussion period is almost halfway over, we kindly inquire if you have any follow-up questions or comments. If we have addressed your concerns, we would really appreciate it if you would consider raising your score.

---

> > > ### Comment · Reviewer_J2jx · 2023-08-16
> > >
> > > I thank the authors for their spirited rebuttal. I especially appreciate the detailed response to my questions. I have read the additional simulation results and the responses to other reviewers' questions. I have decided to raise my rating.

---

> > > > ### Author Response · Authors · 2023-08-16
> > > > **Thank you for the discussion**
> > > >
> > > > Dear Reviewer J2jx,
> > > >
> > > > Thank you for the engaging discussion and willingness to raise your score! We will be sure to include the related literature you mentioned in the next version of the paper, as well as our discussion of ERNIE's limitations.

---

### Official Review · Reviewer_NyS9 · 2023-07-10

**Soundness:** 2 fair
**Presentation:** 2 fair
**Contribution:** 3 good
**Rating:** 5
**Confidence:** 3

**Summary:**

This paper proposes to develop a robust MARL framework based on Lipschitz smoothness of the learned policy (smooth policies). The authors prove the existence of smooth and nearly-optimal policies and propose a framework (ERNIE) to learn robust MARL policy. Experiments are conducted on the two environment: Traffic light control and particle environment.



**Strengths:**

Robust MARL is an important problem to be addressed


**Weaknesses:**

There seems to be many contents squeezed in this paper: theory, framework, connection to game theory (stacklebeg training) and evaluations. It will be nice to have an overview and outline at the end of Intro.

1. This paper also aims to tackle a wide range of threat models if I understand correctly: state perturbation, transition perturbations etc. But why some experiments are evaluating action perturbations?

2. As discussed in Lin 140-141, transition dynamics governed by physical laws (e.g. MuJoCo environment) are of interest. But why in the Experiments, the authors did not show the result of MARL MujoCo Benchmark?

3. Is the ERNIE framework primiarily designed to tackle the threat model of state-perturbation? See Eq 2 of ERNIE regularizer. How can it be used to tackle the transition perturbation threat model?

4. The result in Theorem 3.1 is natural when everything is smooth.


**Questions:**

1. Is the description of Line 48 correct? Is the robustness proprotional to Lipschitz constant or inverse proportional?

2. It is proved that a smooth and close-to-optimal policy exists in smooth enviornment (Line 47-48), but is it guaranteed to obtain such a close-to-optimal policy? and how?

3. Line 215-216 - the boldface and captilaized words does not correspond to ERNIE.

4. Fig 3: the ERNIE's result in Fig 3 seem to be not good, only (a) seem to be outperforming baselines.

5. It's still not clear to me how can the proposed framework be extended beyond smooth environment. Please provide a few example environments that is practically non-smooth and how this framework might be extended?

===================
increase rating from 4 to 5 in the post-rebuttal.

**Limitations:**

The authors have a paragraph discussing potential future work of the method, but not necessarily the limitation.

---

> ### Author Rebuttal · Authors · 2023-08-10
>
> Dear Reviewer NyS9,
>
> Thank you for constructive comments! We believe the weaknesses you pointed out are addressed by our rebuttal. In particular, we make the following changes: (1) Clarify the experimental setting for action robustness (2) Adding new experiments in robotic simulation tasks (3) Clarifying how ERNIE can address the transition perturbation threat.
>
> - **Weakness 0: Include overview and outline at end of introduction.**
> Thank you for this suggestion. We will add an outline/overview to the end of the introduction.
>
> - **Weakness 1: Why are some experiments evaluating action perturbation?**
> In section 4.3 (Line 267), we introduce a version of ERNIE designed to induce robustness against the changing behavior/actions of other agents. To evaluate this version of ERNIE, we conduct experiments where some of the agent's actions are perturbed. We will further clarify the experimental setting and algorithm used in line 336.
>
> - **Weakness 2: Why did we not use the MARL MuJoCo environments?**
> Thank you for the suggestion. To alleviate your concern, we have evaluated ERNIE in the PyBullet (an open-source implementation of MuJoCo) multi-agent drones environment, which also has transition dynamics governed by physical laws. We use this environment over MuJoCo due to compatibility with our hardware. The results can be seen in Figure 3 of the rebuttal material. In this environment ERNIE is shown to be much more robust than the baseline, further confirming the efficacy of ERNIE.
>
> - **Weakness 3: How does ERNIE tackle the transition perturbation threat model?**
> Indeed, ERNIE is primarily designed to tackle the threat model of state-perturbation. However, as we show empirically, ERNIE can also help to tackle the transition perturbation threat model (See Figures 1 and 2). We believe ERNIE provides robustness against transition perturbations due to the data-augmentation that happens when we add adversarially perturbed examples to the training set. Specifically, new scenarios emerge when the environment changes, and data augmentation with adversarial examples provides a large coverage of these scenarios as long as the environment change is small. Adapting to adversarial examples during training ensures that the agents will perform reasonably even in the worst case. Such reasoning can be found in line 61.
>
> - **Weakness 4: Theorem 3.1 holds in smooth environments.**
> It is true that Theorem 3.1 only holds in smooth environments. However, we remark that smoothness is often a good approximation. This can be observed by the good performance of ERNIE in the traffic light control setting, where the reward is not smooth (it is the composition of several categorical functions).
> Furthermore, we remark that theorem 3.3 does not have any smoothness requirement! This suggests that ERNIE can add robustness even in non-smooth environments (see line 191).
>
> - **Question 1: Is lipschitzness proportional to robustness?**
> You are correct here, it should be inversely proportional. We will change it in the next version.
>
> - **Question 2: Are we guaranteed to achieve a close-to-optimal and smooth policy?**
> Currently, we cannot prove convergence to the close to optimal policy. This would be quite difficult due to the difficulty of neural network training and non-stationarity of RL training. However in practice, we find that ERNIE is able to find a policy that is high-performing and fairly smooth (as evidenced by robustness to input perturbations). This indicates ERNIE is an effective way to search for smooth and close to optimal policies.
>
> - **Question 3: ERNIE spelling error.**
> We apologize for this mistake, it will be fixed in the next version.
>
> - **Question 4: Bad performance in Figure 3.**
> Note that ERNIE shows good performance in 3(a) and 3(c). However for 3(b) and 3(d), the baseline policies are already fairly robust (performance doesn’t decrease with increasing perturbation), possibly due to structure of task. Therefore ERNIE cannot show superior performance than the baseline, but performs no worse than the baseline. In essence, we believe that robustness does not matter much on this task, and therefore ERNIE does not show much improvement.
>
> - **Question 5: Extension beyond a smooth environment.**
> There are two ways ERNIE could be extended beyond smooth environments (1) First, ERNIE can straightforwardly be applied to smooth environments with no modification. Although this will result in some approximation errors, the gains in robustness may be worth the added approximation error. An example of this is traffic light control, where the reward function is only partially smooth. This is because the reward is the sum of various categorical variables (i.e. if there was a sudden car stop and the light changed or not). Despite this, ERNIE can still boost performance. This supports our claims that ERNIE can perform well in approximately smooth environments. (2) In addition, we could extend ERNIE to non-smooth environments by using state-adaptive regularization. In particular, we could choose a smaller $\lambda$ in states where the environment is not smooth. We can identify these states by looking at the scale of the ERNIE regularizer (if it is large, then the state is likely non-smooth). We look forward to investigating such a strategy in future work.
>
> - **Limitations:**
> We will add a paragraph on the limitations. In particular, we will discuss the requirement of a smooth or approximately smooth environment, the additional train time of ERNIE, and the fact that ERNIE does not necessarily solve all robustness issues.

---

> > ### Comment · Reviewer_NyS9 · 2023-08-15
> > **Acknowledgement of rebuttal**
> >
> > I thank the authors for the detailed response and additional experiments. I have no other concerns and will increase my rating to 5. The authors please do include the additional experiment results and setup in the camera-ready version, as well as fixing all the inaccurate/incorrect descriptions in the revision to improve the quality of current manuscript.

---

> > > ### Author Response · Authors · 2023-08-15
> > > **Thank you for the discussion**
> > >
> > > Dear Reviewer NyS9,
> > >
> > > Thank you again for the discussion and comments! We will add the additional results to the camera-ready version. We will also fix all inaccurate/incorrect descriptions.

---

### Official Review · Reviewer_mUHD · 2023-07-14

**Soundness:** 3 good
**Presentation:** 3 good
**Contribution:** 3 good
**Rating:** 6
**Confidence:** 4

**Summary:**

This paper proposes a method to train robust policies in a multi-agent reinforcement learning setting via adversarial regularization. The authors showed that robust policies can be achieved by enforcing Lipschitz continuity with respect to states for robustness to noisy states and with respect to actions for noisy action by other potential agents. To enforce the Lipschitz continuity, the authors proposes an adversarial regularization term that can be added to conventional RL training objectives, which minimizes the maximum distance between the policy under a perturbed state/action and the policy under the original state/action. The authors also reformulated the minimax problem into a Stackelberg game, which further stabilize the training process. Finally, the authors also extended their adversarial regularization to the setting of mean-field MARL, which enables the training of large-scale robust marl agents. Experiments and results were provided for several MARL benchmarks.

**Strengths:**

The paper presents a well-written study on a novel approach to train robust multi-agent RL policies. While the core idea of using adversarial regularization may not be entirely novel, its application to multi-agent systems and its extension to mean-field RL is an interesting, albeit expected direction of development. The authors also provided relatively comprehensive experimental results, and the discussions and remarks effectively addressed most of my major questions that I thought of when reviewing the paper. In terms of significance, I believe that this work will further the development of robust MARL systems.


**Weaknesses:**

Overall, I believe the paper could benefit further if the authors could also provide additional experiments in the extension to mean-field MARL where the authors also apply the regularizer to the action to validate their claims for the sake of completeness of experiments. In addition to that, it will also be beneficial if the authors would also provide the run time comparisons for the traffic flow environments on top of the MPE environments for completeness. Finally, if I am understand correctly, both of the experiments are employed a reward structure that is continuous and smooth. It will be more convincing if the authors could also demonstrate the effectiveness of their proposed framework on an environment with an approximately smooth vs non-smooth reward structure.

**Questions:**

I do not have any major questions that requires clarification.

Minor comments/questions:
"Wang et al. [16] consider robustness against uncertain transition dynamics, but their
algorithm is not applied to deep MARL.". This sentence really fall under the subsection of robust RL rather than robust MARL then.

In the supplementary material:
Figure 6: What does STELLA represent? I am assuming its a typo.


**Limitations:**

Yes

---

> ### Author Rebuttal · Authors · 2023-08-09
>
> Dear Reviewer mUHD,
>
> Thank you for the constructive comments! We provide responses to your critiques and questions below.
>
> - **Weakness 1: ERNIE on mean-field actions.**
> Thank you for the suggestion. The results can be seen in Figure 1 of the rebuttal material. We find that applying ERNIE to  mean-field actions does not significantly improve performance. We believe this is the case since we evaluate the algorithm in a cooperative environment, so robustness against other agent actions is not very necessary. As future work, we would like to evaluate ERNIE applied to mean-field actions in competitive (instead of cooperative) environments.
>
> - **Weakness 2: Time comparison on traffic light control.**
> Here are the results: ERNIE takes 0.022 seconds per iteration and the baseline takes 0.023 seconds per iteration.
>
> - **Weakness 3: Evaluation in approximately-smooth environments.**
> We remark that in traffic light control, the reward function is only partially smooth. This is because the reward is the sum of various categorical variables (i.e. if there was a sudden car stop and the light changed or not). Despite this, ERNIE can still boost performance. This supports our claims that ERNIE can perform well in approximately smooth environments.
>
> - **Minor Remark: Discussion of [16].**
> Thank you, we will move the sentence to the appropriate section.
>
> - **Stella typo.**
> Sorry about that, it is a typo! We will fix this in the future versions.

---

> > ### Comment · Reviewer_mUHD · 2023-08-11
> > **Acknowledgement of rebuttal**
> >
> > I would like to thank the authors for the additional results and clarifications and I acknowledge that I have read the rebuttal. I have no further questions

---

### Author Rebuttal · Authors · 2023-08-10

Dear Reviewers,

Thank you for all your helpful comments and critiques! We believe addressing them has significantly improved our work. In particular, we have made several additions to our work:

1. New experiments in multi-agent robotic control.
2. Comparison with new robust MARL algorithms (for both the related work section and the experiments section).
3. Adding new ablation studies (ERNIE for mean-field actions, more time comparisons).
4. Proposing new ways to organize the paper for clarity.

You may find our new experimental results in the attached document. We also provide individual responses to your reviews.

---

### Decision · Program_Chairs · 2023-09-21

**Decision:**

Accept (poster)

**Comment:**

This paper makes a solid contribution to an interesting problem. Some concerns remain, particularly regarding comparisons with existing literature and clarity of the presentation, which the authors are encouraged to address in the final version.